# Study on the Deposition Reduction Effect of the Sediment–Sluice Tunnel in Zengwen Reservoir

**Wei-Cheng Lo [1], Chih-Tsung Huang [1], Meng-Hsuan Wu [1],*, Boris Po-Tsang Chen [2] and Hsi-Nien Tan [1]**

[1] Department of Hydraulic and Ocean Engineering, National Cheng Kung University, No. 1 University Road, Tainan 701, Taiwan
[2] Department of Water Resources Engineering, Feng Chia University, No. 100 Wenhua Road, Taichung 407, Taiwan
* Correspondence: chez_wu@mail.hyd.ncku.edu.tw; Tel.: +886-6-275-7575

**Abstract:** Reservoirs are a crucial part of the human water supply system. The effectiveness and service life of a reservoir is decided mainly by its storage capacity, and as such, preventing reservoir capacity loss is of high interest worldwide. Due to climate change in recent years, precipitation types have changed, and heavy rainfall events have become more severe and frequent. Rainfall causes soil erosion in slope lands and transports large amounts of sediment downstream, forming deposition. This causes reservoir storage capacity to fall rapidly and decreases reservoir service life. The Sediment–Sluice Tunnel can reduce rapid deposition in reservoirs and is, thus, widely employed. By simulating sediment transportation in reservoirs, deposition reduction after building the Sediment–Sluice Tunnel can be evaluated. This study used the Physiographic Soil Erosion–Deposition (PSED) model to simulate the flow discharge and suspended sediment discharge flowing into the Zengwen reservoir then used the depth-averaged two-dimensional bed evolution model to simulate the sediment transportation and deposition in a hydrological process. Simulation results showed that the Sediment–Sluice Tunnel effectively reduced deposition and transported sediment closer to the spillway and Sediment–Sluice Tunnel gate. The deposition distribution with the Sediment–Sluice Tunnel built is more beneficial to the deployment of other dredging works.

**Keywords:** hydraulic flushing; Sediment–Sluice Tunnel; deposition mitigation





## 1. Introduction

Taiwan's water resources come mainly from rivers, groundwater, and reservoirs. According to the 2019 statistics from the Taiwan Water Resources Agency, water taken from rivers, groundwater, and reservoirs amounts to 24.5%, 43.4%, and 32.0% of total usage, respectively. Generally, uses for reservoirs can include water supply, flood control, irrigation, recreation, and power generation [1–3]. Changes to the capacity of a reservoir affect whether the reservoir's designed uses can be fulfilled [4,5].

In recent years, the severity and frequency of both extreme rainfall events and dry events have increased due to climate change [6,7]. This has caused river supply in general to become increasingly unstable and unreliable. Groundwater overuse, on the other hand, causes problems, such as falling groundwater levels, land subsidence, and soil salinization [8,9]. As such, reservoirs have become even more important in terms of adjusting water supply and usage. However, reservoir capacity gradually decreases due to accumulated sediment deposition [10]. According to statistics, global reservoir capacity decreases by approximately 1% annually. This shows that reservoir sedimentation has become a serious problem worldwide [11].

Reservoir sedimentation is a natural phenomenon resulting from upstream sediment entering the reservoir [12,13]. Surface runoff forms when precipitation amounts exceed infiltration capacity and causes flow with high concentrations of sediment from the river

to enter the reservoir. Once inside the reservoir, the increased water depth and surface width slow flow velocity, causing the suspended load and bed load from the river to create deposition on the reservoir bed. This deposition is the main factor in decreased reservoir capacity and can also cause safety concerns, making dredging a necessary component to maintaining reservoir capacity.

Dredging methods often employed in reservoirs include mechanical excavation, deployment of dredging boats, and hydraulic flushing [14–16]. Dredging by mechanical excavation mainly involves excavators entering the reservoir area and mechanically digging out deposited sediment. For mechanical excavation to be implemented, the water level in the reservoir must be low. Dredging via dredging boats involves siphoning water containing high concentrations of suspended sediment from near the reservoir bed and requires water elevation to remain above a specific range while dredging. Hydraulic flushing, on the other hand, uses flood water from rainfall events to drain water with high concentrations of suspended sediment from the reservoir, reducing deposition in the reservoir and maintaining the water and sand balance of the river downstream [17]. As such, hydraulic flushing is the most widely accepted method of reducing reservoir deposition [18].

Not all types of hydraulic flushing are suitable in Taiwan, however. Ways of executing hydraulic flushing include empty flushing, drawdown flushing, and turbidity current flushing [4,10,19]. Empty flushing is performed by using surface runoff or floodwater to scour the reservoir bed and, thus, requires the reservoir to be emptied beforehand. Due to this condition, the reservoir cannot fulfill its original purposes, namely water supply and flood control, during empty flushing. Drawdown flushing is performed by discharging water from the reservoir. Scour forces produced while lowering the reservoir's water level flush out sediment near the outlet. Turbidity current flushing uses mud water's characteristic of being heavier than freshwater to discharge only the suspended sediment-filled water under the freshwater in the reservoir from the outlet. Both empty flushing and drawdown flushing require the lowering of water levels during the flood season and increase the risk of water shortage and drought during the dry season. With drought risk during the dry season increasing in recent years [20,21], methods that can retain more water storage are generally preferred. The turbidity current flushing method has both the advantages of reducing reservoir sedimentation and reusing more freshwater reserves and is, thus, considered a better choice for reservoirs in Taiwan. Numerical models are required to simulate sediment evacuation processes to research the effects of turbidity current flushing. Numerous numerical models have been applied to the simulation of reservoir sediment transportation in the past. For example, Wang et al. used the SRH-2D model to simulate drawdown flushing in the Agongdian Reservoir in Taiwan during Typhoon Talim [10]. Dutta and Sen used the TELEMAC-2D model to simulate relations between sedimentation, storage capacity, and operational life of the Hirakud Reservoir in India [22]. Moussa used the CCHE-2D model to simulate sediment transportation and estimate the effective life span of the Aswan High Dam Reservoir in Egypt [23]. This study chose the Zengwen reservoir, the largest reservoir in Taiwan, as the study area for the simulation of turbidity current flushing. The PSED model and the depth-averaged two-dimensional bed evolution model were used to simulate the sediment yield of the reservoir watershed and sediment erosion and deposition in the reservoir. Simulation results were used to analyze the effects of the Sediment–Sluice Tunnel and to possibly provide a future reference for the reservoir management agency in the field of dredging.

## 2. Materials and Methods

### 2.1. Study Area

This study selected the Zengwen Reservoir in Southwestern Taiwan as the study area. The Zengwen Reservoir is located in the upstream area of the Zengwen River and has a watershed area of 481 square kilometers. The elevation of the watershed area is between 180 and 2600 m, with an average of approximately 1000 m. The distribution of elevation

in the Zengwen reservoir is shown in Figure 1. According to sediment sampling data, the sediment median diameter ($D_{50}$) of the Zengwen reservoir bed is 21.45 μm [24]. Size graduation of sediment in the Zengwen reservoir is shown in Table 1.

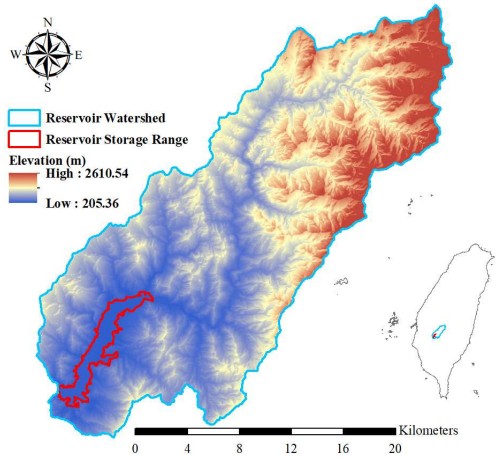

**Figure 1.** Elevation of the Zengwen Reservoir.

**Table 1.** Particle size distribution of sediment in the Zengwen Reservoir.

| $D_{10}$ | $D_{25}$ | $D_{50}$ | $D_{75}$ | $D_{90}$ |
|---|---|---|---|---|
| 2.92 | 7.69 | 21.45 | 43.51 | 70.99 |

Units: μm.

The Zengwen Reservoir, also called Tsengwen Reservoir, was built in 1973, with a design total capacity of 748.4 million cubic meters, and is currently the reservoir with the largest capacity in Taiwan [5]. With the number of extreme rainfall events increasing due to climate change, sediment deposition amounts in the reservoir have increased. According to the reservoir sedimentation measurement of the Zengwen Reservoir, the average sediment deposition amount per year from 1973 to 2000 is 3.94 million cubic meters. This increased to an average of 10.06 million cubic meters per year during the 2001 to 2008 period. Typhoon Morakot, the worst natural disaster in recent Taiwan, occurred in 2009. It caused a disastrous amount of 91.08 million cubic meters of sediment to be deposited into the Zengwen Reservoir [25]. After Typhoon Morakot, large areas of the Zengwen Reservoir watershed collapsed and caused the average yearly sediment deposition amount from 2010 to 2017 to be greater than that of 1973 to 2000, at an average of 4.59 million cubic meters yearly. The period 2018 to 2020 saw a decrease in sediment deposition amounts, however, due to less precipitation during the period. The Southern Region Water Resources Office, the agency in charge of reservoir management, also actively performed dredging in the area during the period mentioned above [26].

The Ministry of Economic Affairs of Taiwan modified reservoir operation directions in 2018, changing the water storage elevation from the elevation of 227 m to 230 m and increasing reservoir capacity by 55 million cubic meters as a result [27]. Yearly sediment deposition and changes in Zengwen Reservoir's capacity are shown in Figure 2. According to surveying in 2020, reservoir capacity has decreased by 36.4% of its original size since 1973.

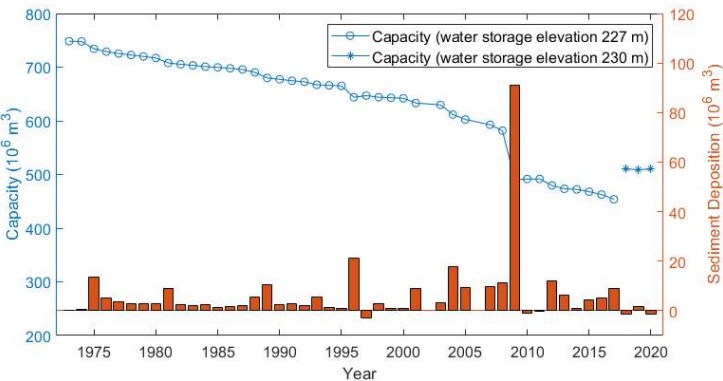

**Figure 2.** Sediment deposition amounts per year in the Zengwen Reservoir.

## 2.2. Hydraulic Structures

The Zengwen reservoir has been in operation for more than half a century. The original design of the reservoir included a spillway and Permanent River Outlet (PRO). Outlet gates of the spillway have a maximum flow rate of 11,345 cm total, and the PRO has a maximum flow rate of 180 cm. According to the Operation Directions for Gates of Tsengwen Reservoir, both types of outlets are normally closed unless flood control or maintenance is required.

With the increase of sediment deposition in the reservoir, the reservoir's water storage ability and service life were affected. This was worsened by the 2009 Morakot Typhoon, as the typhoon resulted in sediment deposition reaching elevations of 176 m, exceeding that of PRO inlets by 20 m and drastically affecting PRO operation. The Southern Region Water Resources Office built a 1.2 km long Sediment–Sluice Tunnel from 2014 to 2017 to maintain functionality and extend the expected service life of the reservoir [28]. The maximum flow rate of the Sediment–Sluice Tunnel is 1070 cm and was expected to remove 1.04 million cubic meters of sediment from the reservoir annually. The positioning of Zengwen reservoir's facilities is shown in Figure 3.

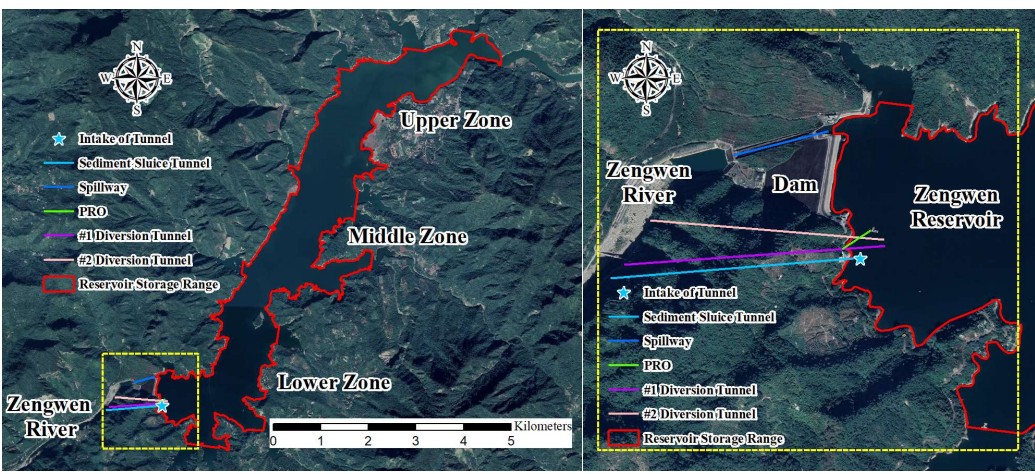

**Figure 3.** Location of hydraulic structures of the Zengwen Reservoir.

## 2.3. Numerical Model

This study used the PSED model to simulate the flow discharge and suspended sediment discharge of the Zengwen Reservoir watershed. The results were used as the upstream boundary condition for the depth-averaged two-dimensional bed evolution model while simulating sediment concentration transportation and the reservoir bed deposition process. Finally, the simulated sediment concentration transportation and the reservoir bed deposition process were used to discuss the effects of the Sediment–Sluice Tunnel on sediment transportation in the reservoir.

2.3.1. Physiographic Soil Erosion–Deposition Model

The PSED model can be divided into two parts, rainfall-runoff simulation and soil erosion–deposition simulation [4]. The government equation for the rainfall-runoff simulation is as follows [29]:

$$A_i \frac{dh_i}{dt} = Pe_i + \sum_k Q_{i,k}(h_i, h_k) \tag{1}$$

where $A_i$ is the area of cell $i$; $h_i$ and $h_k$ express the water level of cell $i$ and neighboring cell $k$, respectively, at time $t$; $Q_{i,k}$ denotes the discharge from cell $k$ to its neighboring cell $i$; and $Pe_i$ represents the effective rainfall volume per unit time $t$ in cell $i$, which is equal to the effective rainfall per unit time $t$ in cell $i$ multiplied by its area $A_i$.

Using the explicit finite difference method, Equation (1) can be expressed as follows:

$$h_i^{m+1} = h_i^m + \frac{\left( \sum_k Q_{i,k}^m + P_{ei}^m \right)}{A_i} \cdot \Delta t \tag{2}$$

$$D_i = h_i - z_i \tag{3}$$

where the superscript $m$ denotes the known physical quantity at time $t_m$; $m + 1$ denotes the unknown physical quantity at time $t_{m+1}$; $\Delta t$ is the time interval of calculation; $D_i$ denotes the water depth of cell $i$; and $z_i$ is the bed elevation of cell $i$.

The government equation for soil erosion–deposition can be expressed as follows [4,13,30,31]:

$$\frac{\partial V_{si}}{\partial t} = \sum_k Q_{SC_{i,k}} + Q_{sei} - Q_{sdi} + R_{DTi} \tag{4}$$

$$(1 - \lambda) \frac{\partial V_{di}}{\partial t} = \sum_k Q_{SB_{i,k}} - Q_{sei} + Q_{sdi} - R_{DTi} \tag{5}$$

where $V_{si}$ represents the volume of suspended sediment load in the water body of cell $i$ and is equal to the area $A_i$ multiplied by the water depth $D_i$ multiplied by the suspended sediment concentration $C_i$; $V_{di}$ denotes the volume of alluvium in cell $i$; $\lambda$ expresses the porosity of bed material; $Q_{SC_{i,k}}$ and $Q_{SB_{i,k}}$ are the discharge of suspended load and discharge of bed load from cell $k$ to cell $i$, respectively; $Q_{sei}$ denotes the entrainment rate of the riverbed or land surface of cell $i$; $Q_{sdi}$ represents the deposition rate of cell $i$; and $R_{DTi}$ indicates the rainfall detachment rate of cell $i$.

Using the explicit finite difference method, Equations (4) and (5) can be expressed as follows:

$$C_i^{m+1} = \frac{\left( \sum_k Q_{SC_{i,k}}^m + Q_{sei}^m - Q_{sdi}^m + R_{DTi}^m \right)}{A_i D_i^{m+1}} \cdot \Delta t \tag{6}$$

$$\frac{\Delta V_{di}}{A_i} = \Delta z_i = \frac{\left( \sum_k Q_{SB_{i,k}}^m - Q_{sei}^m + Q_{sdi}^m - R_{DTi}^m \right)}{A_i(1 - \lambda)} \cdot \Delta t \tag{7}$$

$$z_i^{m+1} = z_i^m + \Delta z_i \tag{8}$$

This study built a total of 4407 computational cells on the basis of the hydrological and physiographic data of the study area, with cell area ranging from 0.41 to 58.04 hectares, as shown in Figure 4.

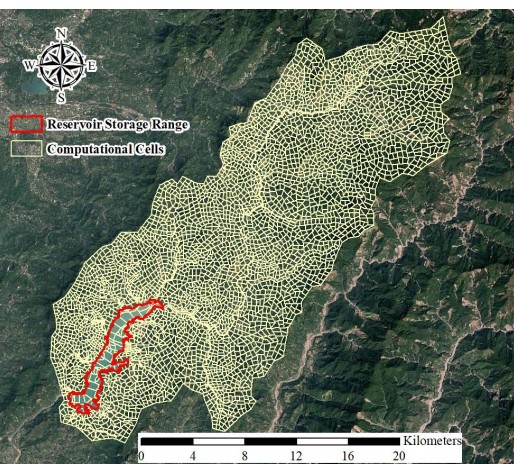

**Figure 4.** Computational cells of the Zengwen Reservoir watershed.

2.3.2. The Depth-Averaged Two-Dimensional Bed Evolution Model

The depth-averaged two-dimensional bed evolution model is used mainly to simulate discharge and sediment transportation over time in the reservoir. The depth-average continuity and momentum equations of flow can be expressed as follows [32]:

$$\frac{\partial h}{\partial t} + \frac{\partial (h\overline{u})}{\partial x} + \frac{\partial (h\overline{v})}{\partial y} = 0 \tag{9}$$

$$\frac{\partial (\overline{u}h)}{\partial t} + \frac{\partial (\overline{u}\,\overline{u}h)}{\partial x} + \frac{\partial (\overline{u}\,\overline{v}h)}{\partial y} = \frac{\partial}{\partial x}\left(2\overline{\varepsilon_{xx}}h\frac{\partial \overline{u}}{\partial x}\right) + \frac{\partial}{\partial y}\left[\overline{\varepsilon_{xy}}h\left(\frac{\partial \overline{u}}{\partial y} + \frac{\partial \overline{v}}{\partial x}\right)\right] - \frac{\tau_{bx}}{\rho} - gh\frac{\partial H}{\partial x} \tag{10}$$

$$\frac{\partial (\overline{v}h)}{\partial t} + \frac{\partial (\overline{v}\,\overline{u}h)}{\partial x} + \frac{\partial (\overline{v}\,\overline{v}h)}{\partial y} = \frac{\partial}{\partial x}\left[\overline{\varepsilon_{xy}}h\left(\frac{\partial \overline{u}}{\partial y} + \frac{\partial \overline{v}}{\partial x}\right)\right] + \frac{\partial}{\partial y}\left(2\overline{\varepsilon_{yy}}h\frac{\partial \overline{v}}{\partial y}\right) - \frac{\tau_{by}}{\rho} - gh\frac{\partial H}{\partial y} \tag{11}$$

where $h$ is the water depth; $H$ denotes the water surface elevation; $\overline{u}$ and $\overline{v}$ represents the depth-average flow velocity components in $x$ and $y$ directions, respectively; $\overline{\varepsilon_{xx}}$, $\overline{\varepsilon_{xy}}$, and $\overline{\varepsilon_{yy}}$ are the depth-average kinematic eddy viscosities of flow; $\rho$ is density of flow; $g$ is gravitational constant; and $\tau_{bx}$ and $\tau_{by}$ indicate bed shear stresses $\tau_b$ in $x$ and $y$ directions, respectively.

Using the explicit finite difference method, Equation (9) can be expressed as follows:

$$h_{i,j}^{m+1} = h_{i,j}^{m} + \Omega_h(h^m, \overline{u}^m, \overline{v}^m) \tag{12}$$

where $\Omega_h$ is the known function composed of $h$, $\overline{u}$, and $\overline{v}$ at time $t_m$.

The depth-average convective-diffusive equation of suspended sediment can be expressed as follows [14,15]:

$$\frac{\partial (\overline{C}h)}{\partial t} + \frac{\partial (\overline{u}\overline{C}h)}{\partial x} + \frac{\partial (\overline{v}\overline{C}h)}{\partial y} = \frac{\partial}{\partial x}\left(\overline{E_x}\frac{\partial \overline{C}}{\partial x}\right) + \frac{\partial}{\partial y}\left(\overline{E_y}\frac{\partial \overline{C}}{\partial y}\right) + q_{se} - q_{sd} \tag{13}$$

$$\frac{\partial Z}{\partial t} + \frac{1}{1-\lambda}\left[\frac{\partial q_{bx}}{\partial x} + \frac{\partial q_{by}}{\partial y} + (q_{se} - q_{sd})\right] = 0 \tag{14}$$

where $\overline{C}$ is the depth-averaged volumetric concentration of suspended load; $\overline{E_x}$ and $\overline{E_y}$ are the sediment mass diffusivity; $Z$ denotes the bed elevation; and $q_{bx}$ and $q_{by}$ represent the bed load transport rate per unit width $q_b$ in the direction of $x$ and $y$, respectively.

Water discharge scouring the bed causes bed load. The bed load transportation rate per unit width $q_b$ can be calculated by the Meyer-Peter and Müller equation shown as follows [32]:

$$\left(\frac{k_n}{k'}\right)^{\frac{3}{2}} \gamma h S_f = 0.047(\gamma_s - \gamma)d_m + 0.25\left(\frac{\gamma}{g}\right)^{\frac{1}{3}}\left(\frac{\gamma_s - \gamma}{\gamma}\right)^{\frac{2}{3}} q_b^{\frac{2}{3}} \tag{15}$$

$$k_n = \frac{1}{n} \tag{16}$$

$$k' = \frac{26}{d_{90}^{1/6}} \tag{17}$$

where $\gamma$ is the specific weight of water; $\gamma_s$ denotes the specific weight of sediment; $S_f$ is the friction slope; $n$ is the manning coefficient; $d_m$ is average particle size; and $d_{90}$ expresses the size diameter coarser than 90% of the grains in the sample.

Using the explicit finite difference method, Equations (13) and (14) can be expressed as follows:

$$\overline{C}_{i,j}^{m+1} = \frac{\overline{C}_{i,j}^m h_{i,j}^m}{h_{i,j}^{m+1}} + \Omega_c\left(h^m, \overline{u}^m, \overline{v}^m, \overline{C}^m\right) \tag{18}$$

$$Z_{i,j}^n = Z_{i,j}^n + \Omega_z\left(h^m, \overline{u}^m, \overline{v}^m, \overline{C}^m\right) \tag{19}$$

where $\Omega_c$ and $\Omega_z$ are the known functions composed of $h$, $\overline{u}$, $\overline{v}$, and $\overline{C}$ at time $t_m$.

The depth-averaged two-dimensional bed evolution model used the result of deposition measurement of the reservoir from 2017 and used 10 m × 10 m regular grids to divide the reservoir into 131,432 cells.

## 3. Results and Discussion

### 3.1. Analysis of Sediment Yield

As of the end of 2021, the Sediment–Sluice Tunnel of Zengwen Reservoir had experienced eight typhoon rainfall events since its completion in November 2017. The 0823 rainfall event of 2018 was chosen from the eight events mentioned above because it had the largest estimated amount of sediment flowing into the reservoir. During the event, the amount of sediment deposition that flowed into the reservoir from the mainstream alone amounted to 2.22 million cubic meters [33]. This study simulated sediment concentration distribution from 04:00, 23 August to 00:00, 26 August, with the results shown in Figure 5. The first peak flow discharge of the hydrological process occurred 14 h into the simulation (18:00, 23 August), had a discharge rate of 2186 cm, and was also the largest peak flow discharge of the hydrological process. The first peak suspended sediment discharge occurred 16 h into the simulation, with a discharge rate of 32.56 cm, while the largest peak suspended sediment discharge occurred 23 h into the simulation, with a discharge rate of 34.59 cm. According to simulation results, the mainstream and tributary resulted in a total of 3.25 million cubic meters of sediment deposition subsiding in the reservoir. The mainstream in the simulation deposited 2.42 million cubic meters of sediment into the reservoir, a 9% increase from what was actually observed by the Southern Region Water Resources Office. This shows that the PSED model can effectively estimate water flow and sediment deposition into reservoirs.

### 3.2. Analysis of Sediment Transportation in the Reservoir

To understand sediment transportation in the reservoir, the flow discharge and suspended sediment discharge was simulated by the PSED model, and the results were used as the boundary condition of the depth-averaged two-dimensional bed evolution model. The simulated concentration of suspended sediment over time is shown in Figure 6. As seen in the figure, suspended sediment started to enter the upper zone of the reservoir 12 h into the simulation, with concentrations ranging from 1000 to 3000 ppm. At 24 h into the simulation, with sediment from upstream entering the reservoir area, sediment concentra-

tions gradually rose, reaching a maximum of 20,000 ppm while being transported toward the lower zone. There was also a high turbidity current entering from the west during that time, with sediment concentrations in the high turbidity current reaching 50,000 ppm. The high turbidity flow originated from the bare lands adjacent to the reservoir's west tributary, an area prone to high turbidity flows due to also being in a landslide geologically sensitive area. At 36 h into the simulation, suspended sediment had reached the reservoir's lower zone and had started exiting the reservoir via the spillway, PRO, and Sediment–Sluice Tunnel. At 48 h into the simulation, concentrations of suspended sediment in the high turbidity current from the tributary remained high, but concentrations of suspended sediment in the lower zone had begun to decrease. At 60 h into the simulation, the flow discharge and suspended sediment discharge of the mainstream had decreased, and concentrations in the reservoir dropped below 5000 ppm in almost all areas of the reservoir. Another high turbidity current entered the reservoir toward the end of the simulation, however, and caused regional suspended sediment concentrations to rise.

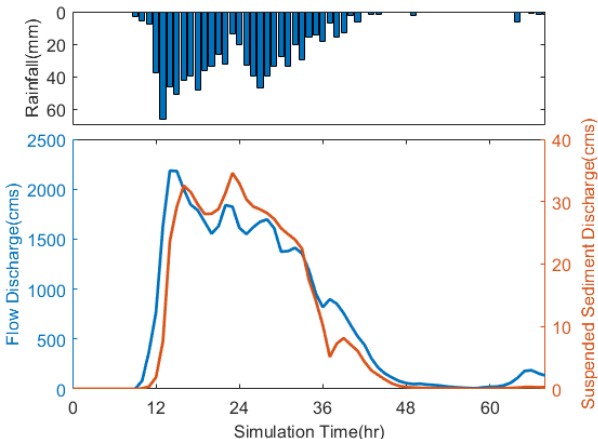

**Figure 5.** Simulated flow discharge and suspended sediment discharge during the 0823 rainfall event.

### 3.3. Analysis of Erosion and Deposition in the Reservoir

The distribution of sediment concentration is highly correlated with the distribution of sediment deposition already in the reservoir. The simulation of the 0823 rainfall event flushed a total of 0.36 million cubic meters of sediment from the reservoir, an increase of 0.02 million cubic meters from the actual amount observed by the reservoir management agency [33]. This showed that the PSED model could effectively estimate the amount of flushed sediment. The simulated changes to sediment distribution are shown in Figure 7. At 12 h into the simulation, with the sediment just arriving at the reservoir, bed elevation in the reservoir had not changed by evident margins. At 24 h into the simulation, the upper zone and the regions adjacent to the west tributary saw sparse deposition of approximately 0.2 m in height. With flow velocity in the reservoir decreasing along with the decrease in discharge into the reservoir, large deposition areas occurred 36 h into the simulation. The distribution of sediment deposition was similar to that of sediment concentrations, with deposition height ranging from 0.2 to 0.4 m. Sediment deposition then increased in size and height as time passed. At 60 h into the simulation, deposit areas had reached the spillway of the reservoir. At the end of the simulation, the upper zone of the reservoir had deposition height ranging from 0.4 to 0.6 m. On the other hand, the middle zone of the reservoir was the most severely affected zone due to the high turbidity current that entered the reservoir, with deposition height mostly within the 0.8 to 1.2 m range. However, parts of the middle zone had 1.8 to 2.0 m high deposition.

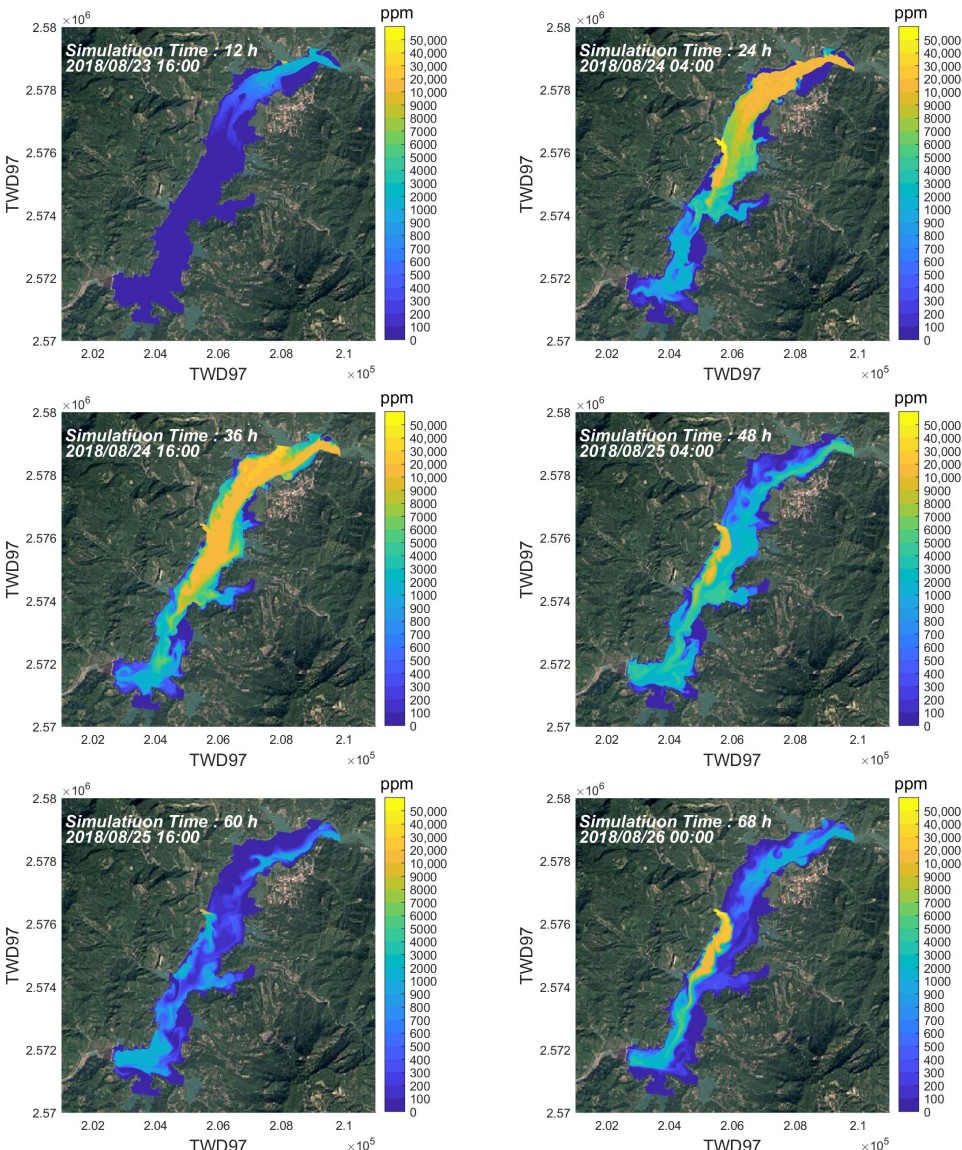

**Figure 6.** Simulated suspended sediment concentration distribution in the Zengwen Reservoir during the 0823 rainfall event (with the Sediment–Sluice Tunnel).

*3.4. Discussion on the Effects of the Sediment–Sluice Tunnel on Reservoir Sedimentation*

To discuss the impact building the Sediment–Sluice Tunnel has on sediment deposition in reservoirs, this study re-simulated the 0823 rainfall event with the Sediment–Sluice Tunnel closed to simulate the scenario had the 0823 rainfall event occurred in the Zengwen reservoir without the Sediment–Sluice Tunnel. The changes to sediment deposition distribution are shown in Figure 8. Simulations showed that sediment deposition occurred mainly in the upstream and middle regions of the reservoir, and increases in deposition amount were observed. The comparison of sediment deposition height distribution with and without the Sediment–Sluice Tunnel is shown in Figure 9, with red areas in the figure showing the regions with increased sediment deposition height after the Sediment–Sluice Tunnel was built, and blue areas showing the regions with decreased sediment deposition height. It can be seen from the simulation results that the Sediment–Sluice Tunnel had less impact on the northern regions of the reservoir. The middle regions of the reservoir had decreases of 0.2 to 0.6 m in sediment deposition height, and regions close to the spillway had increases ranging from 0 to 0.2 m, when compared with the situation without the Sediment–Sluice Tunnel.

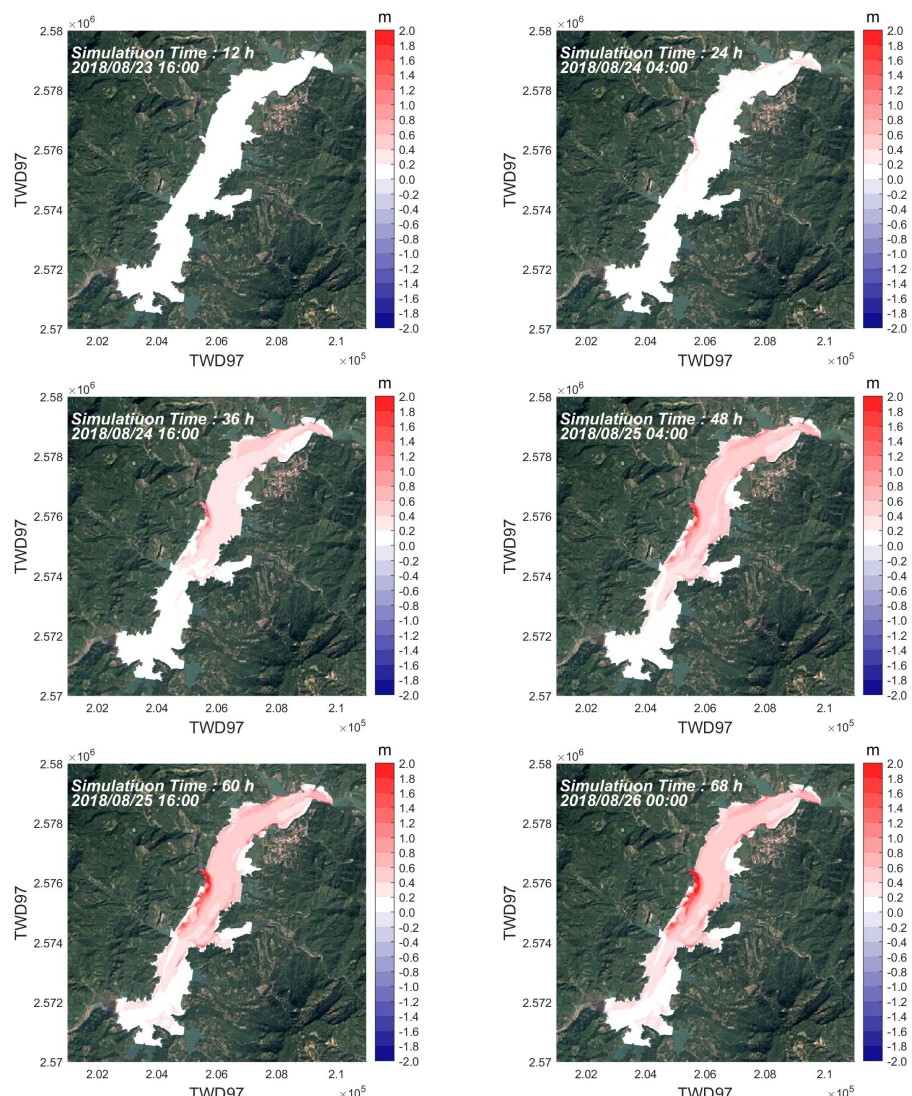

**Figure 7.** Simulated erosion–deposition distribution in the Zengwen Reservoir during the 0823 rainfall event (with the Sediment–Sluice Tunnel).

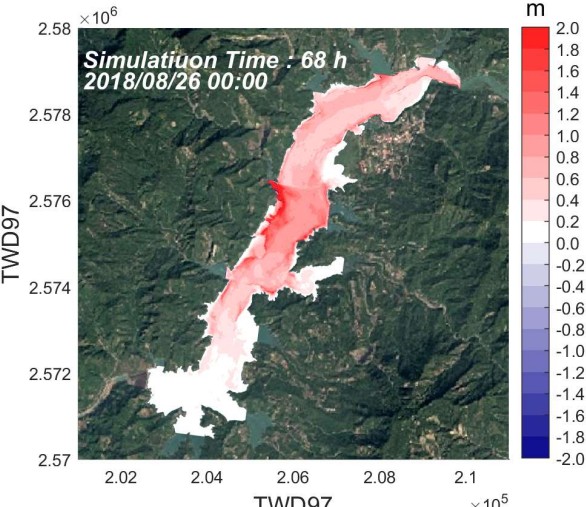

**Figure 8.** Simulated erosion–deposition distribution in the Zengwen Reservoir during the 0823 rainfall event (without the Sediment–Sluice Tunnel).

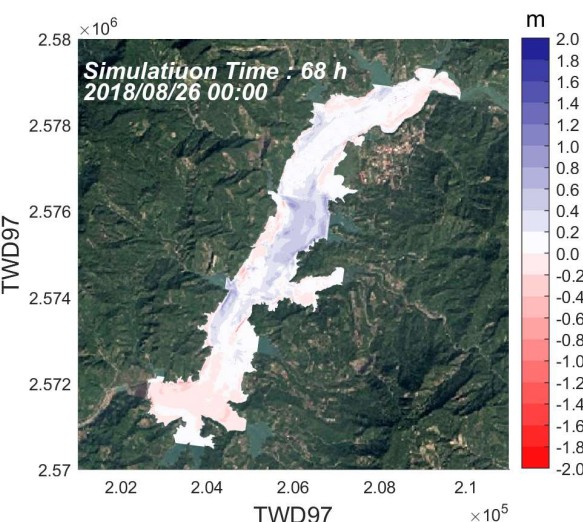

**Figure 9.** Comparison of simulated deposition distribution with and without the Sediment–Sluice Tunnel in the Zengwen Reservoir.

## 4. Conclusions

This study aims mainly to investigate the effects of the Zengwen Sediment–Sluice Tunnel on the sediment transportation phenomenon by simulating flow discharge and suspended sediment discharge with the PSED model and the distribution of sediment concentration and deposition with the depth-averaged two-dimensional bed evolution model. This study chose the 0823 rainfall event as the simulation case and simulated the differences in sediment deposition height and distribution.

In the simulation with the Sediment–Sluice Tunnel, the reservoir's middle zone had the most severe sediment deposition, with most areas in the zone having 0.8 to 1.2 m high deposition, and a few areas having 1.8 to 2.0 m high deposition. The upper and lower zones of the reservoir had less sediment deposition, with heights ranging from 0.2 to 0.6 m and 0 to 0.2 m, respectively. When compared with the simulation without the Sediment–Sluice Tunnel, it can be seen that deposition area and height decreased noticeably in the upper and middle zones, while increasing in the lower zone after the Sediment–Sluice Tunnel was built. Sediment deposition reduction was especially profound in the middle zone, with deposition height decreasing by 0.4 to 0.6 m after the Sediment–Sluice Tunnel was built. Large amounts of sediment were transported toward the lower zone, causing regional deposition height to increase by 0 to 0.2 m.

The Sediment–Sluice Tunnel can effectively decrease sedimentation in the Zengwen Reservoir and can transport sediment inside the reservoir closer to the lower zone of the reservoir. However, due to the absence of rainfall events in the Zengwen Reservoir watershed scouring large amounts of sediment into the Zengwen Reservoir after the 0823 event, the Sediment–Sluice Tunnel could not reach its full potential in recent years. In similar situations, dredging boats or other dredging methods are recommended to extend reservoirs' service life expectancy.

**Author Contributions:** Conceptualization, M.-H.W. and B.P.-T.C.; methodology, C.-T.H. and M.-H.W.; validation, C.-T.H. and M.-H.W.; formal analysis, C.-T.H., M.-H.W., and H.-N.T.; data curation, H.-N.T.; writing—original draft preparation, C.-T.H. and M.-H.W.; writing—review and editing, C.-T.H. and M.-H.W.; visualization, C.-T.H.; supervision, W.-C.L.; project administration, B.P.-T.C.; All authors have read and agreed to the published version of the manuscript.

**Funding:** This research was funded by the Ministry of Science and Technology (MOST) of Taiwan, grant number MOST107-2119-M035-003.

**Institutional Review Board Statement:** Not applicable.

**Informed Consent Statement:** Not applicable.

**Data Availability Statement:** Data are available from the corresponding author.

**Acknowledgments:** The authors appreciate Southern Region Water Resources Office, Water Resources Agency, Taiwan, and Central Weather Bureau, Taiwan for providing the important information and data used in this paper.

**Conflicts of Interest:** The authors declare no conflict of interest.

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
