# Peer review of "Study on the Deposition Reduction Effect of the Sediment–Sluice Tunnel in Zengwen Reservoir"

_water, doi:10.3390/w15061072_

Round 1

Reviewer 1 Report

This manuscript investigates the deposition reduction effect of the Sediment Sluice Tunnel in Zengwen Reservoir in Taiwan, China using numerical simulations. This study is scientifically meaningful and may lead to potential engineering applications. The manuscript is well organized and well written in general in my opinion. However, there are still some important issues to be addressed.

1 The descriptions on the mathematical models are insufficient. For example, how to calculate the sediment transport rate? What’s the governing equations for bedload transport if bedload is involved? How the governing equations are discretized and solved numerically?

2 Usually numerical simulation results shall be verified in some way. However, there is no any verifications in the manuscript of the current version.

3 Research progress on numerical simulations with similar methods had better be discussed in the introduction.

Author Response

  1. The descriptions on the mathematical models are insufficient. For example, how to calculate the sediment transport rate? What’s the governing equations for bedload transport if bedload is involved? How the governing equations are discretized and solved numerically?
  • Thank you for your suggestion, the governing equations and related descriptions have been added to the manuscript.

  1. Usually numerical simulation results shall be verified in some way. However, there is no any verifications in the manuscript of the current version.
  • Thank you for your suggestion. According to the reservoir sedimentation measurements of the Zengwen Reservoir, the 0823 rainfall event carried the largest estimated amount of suspended sediment of all recent rainfall events. Simulated results of the 0823 rainfall event had a 0.2 million cubic meter difference from the observed results, and the simulated amount of sediment flushed from the reservoir had a 0.02 million cubic difference from the actual results; This study believes that both differences can be considered reasonable. In addition, the Zengwen Reservoir watershed has no water discharge stations, and thus water discharge could not be compared.
  1. Research progress on numerical simulations with similar methods had better be discussed in the introduction.
  • Thank you for your suggestion, we have added examples of numerical models used in the past for the simulation of sediment transportation in reservoirs to the manuscript.

Reviewer 2 Report

This paper is well written and close to ready for publication. I have a few small suggestions that should not take too long.

(1) First it would help to have a better figure of the dam showing the location of tunnel intakes and the spillway. 

(2) The authors claim that the sole cause of the multi-decadal rise in sediment flux to the reservoir is the climate-change effects on precipitation intensity. Is there any impact of human development in the watershed?

(3) Provide a little bit more detail on the turbidity flush method to preferentially get sediment out of the reservoir bed without passing a lot of water.

(4) The concentrations of sediment that occur as a result of major runoff events are extraordinary, up to 50,000 ppm. This is well into the fluid mud range and that of density flows. How comfortable are you using a 2D model to predict such transport and deposition? Do you have any recommendations?

(5) I presume the sediment that is coming into the reservoir is fine-grained but the paper provides no information about the size gradation of the sediment in flux.

(6) The highest sediment concentrations enter the reservoir from a small tributary on the west side. Why does it produce so much suspended sediment flux?

Author Response

  1. First it would help to have a better figure of the dam showing the location of tunnel intakes and the spillway.
  • Thank you for your suggestion, we have marked the location of tunnel intakes and the spillway in the figure.
  1. The authors claim that the sole cause of the multi-decadal rise in sediment flux to the reservoir is the climate-change effects on precipitation intensity. Is there any impact of human development in the watershed?
  • The Zengwen reservoir watershed is currently composed mainly of woodlands and slopeland conservation areas, with few settlements in the surrounding areas. We believe that the impact of human development in the watershed can be neglected.
  1. Provide a little bit more detail on the turbidity flush method to preferentially get sediment out of the reservoir bed without passing a lot of water.
  • Thank you for your suggestion, we have added numerical turbidity flush method related descriptions to the manuscript.
  1. The concentrations of sediment that occur as a result of major runoff events are extraordinary, up to 50,000 ppm. This is well into the fluid mud range and that of density flows. How comfortable are you using a 2D model to predict such transport and deposition? Do you have any recommendations?
  • According to simulation results, if the high turbidity flow with approximately 50000 ppm of suspended sediment from the west is disregarded, suspended sediment concentration in the reservoir had a peak value of approximately 20000 ppm. This value decreased along with the concentration in the mainstream, and most regions in the reservoir observed values of approximately 5000 ppm thereafter. It is still possible to use the 2D model to simulate transportation and deposition under these conditions.
  1. I presume the sediment that is coming into the reservoir is fine-grained but the paper provides no information about the size gradation of the sediment in flux.
  • Thank you for your suggestion, we have added the size gradation of the sediment in flux to the manuscript.
  1. The highest sediment concentrations enter the reservoir from a small tributary on the west side. Why does it produce so much suspended sediment flux?
  • The Zengwen Reservoir watershed is situated in a landslide geologically sensitive area and is thus prone to natural disasters such as landslide and debris flow. The fact that the area surrounding the west tributary is mainly composed of bare land only increases the odds of high turbidity flows occurring.

Round 2

Reviewer 1 Report

The authors have well answered all of my question and made revisions in manuscript accordingly. I have no further comments.